

# Improving taxonomic classification of marine zooplankton by molecular approach: registration of taxonomically verified 18S and 28S rRNA gene sequences

Tsuyoshi Watanabe[1], Junya Hirai[2], Sirje Sildever[3,4], Kazuaki Tadokoro[5], Kiyotaka Hidaka[3], Iwao Tanita[6], Koh Nishiuchi[7], Naoki Iguchi[8], Hiromi Kasai[1], Noriko Nishi[9], Seiji Katakura[10], Yukiko Taniuchi[1], Taketoshi Kodama[3,12], Satokuni Tashiro[6], Misato Nakae[8], Yuji Okazaki[5], Satoshi Kitajima[7], Sayaka Sogawa[3], Toru Hasegawa[7], Tomonori Azumaya[1], Yutaka Hiroe[7], Daisuke Ambe[3], Takashi Setou[3], Daiki Ito[3], Akira Kusaka[3], Takeshi Okunishi[5], Takahiro Tanaka[5], Akira Kuwata[5], Daisuke Hasegawa[5], Shigeho Kakehi[5], Yugo Shimizu[5] and Satoshi Nagai[11]

[1] Kushiro Field Station, Fisheries Resources Institute, Japan Fisheries Research and Education Agency, Kushiro, Japan
[2] Atmosphere and Ocean Research Institute, The University of Tokyo, Kashiwa, Japan
[3] Yokohama Field Station, Fisheries Resources Institute, Japan Fisheries Research and Education Agency, Shiogama, Japan
[4] Department of Marine Systems, Tallinn University of Technology, Tallinn, Estonia
[5] Shiogama Field Station, Fisheries Resources Institute, Japan Fisheries Research and Education Agency, Shiogama, Japan
[6] Yaeyama Field Station, Fisheries Technology Institute, Japan Fisheries Research and Education Agency, Ishigaki, Japan
[7] Nagasaki Field Station, Fisheries Resources Institute, Japan Fisheries Research and Education Agency, Nagasaki, Japan
[8] Niigata Field Station, Fisheries Resources Institute, Japan Fisheries Research and Education Agency, Niigata, Japan
[9] AXIOHELIX Co. Ltd., Tokyo, Japan
[10] City of Mombetsu, Mombetsu, Japan
[11] Yokohama Field Station, Fisheries Technology Institute, Fisheries Research and Education Agency, Yokohama, Japan
[12] Present Address: Graduate School of Agricultural and Life Sciences, The University of Tokyo, Japan

Corresponding author
Tsuyoshi Watanabe,
watanabe_tsuyoshi25@fra.go.jp

## ABSTRACT

**Background.** Zooplankton plays an important role in the marine ecosystem. A high level of taxonomic expertise is necessary for accurate species identification based on morphological characteristics. As an alternative method to morphological classification, we focused on a molecular approach using 18S and 28S ribosomal RNA (rRNA) gene sequences. This study investigates how the accuracy of species identification by metabarcoding improves when taxonomically verified sequences of dominant zooplankton species are added to the public database. The improvement was tested by using natural zooplankton samples.

**Methods.** rRNA gene sequences were obtained from dominant zooplankton species from six sea areas around Japan and registered in the public database for improving the

accuracy of taxonomic classifications. Two reference databases with and without newly registered sequences were created. Comparison of detected OTUs associated with single species between the two references was done using field-collected zooplankton samples from the Sea of Okhotsk for metabarcoding analysis to verify whether or not the newly registered sequences improved the accuracy of taxonomic classifications.

**Results**. A total of 166 sequences in 96 species based on the 18S marker and 165 sequences in 95 species based on the 28S marker belonging to Arthropoda (mostly Copepoda) and Chaetognatha were registered in the public database. The newly registered sequences were mainly composed of small non-calanoid copepods, such as species belonging to *Oithona* and *Oncaea*. Based on the metabarcoding analysis of field samples, a total of 18 out of 92 OTUs were identified at the species level based on newly registered sequences in the data obtained by the 18S marker. Based on the 28S marker, 42 out of 89 OTUs were classified at the species level based on taxonomically verified sequences. Thanks to the newly registered sequences, the number of OTUs associated with a single species based on the 18S marker increased by 16% in total and by 10% per sample. Based on the 28S marker, the number of OTUs associated with a single species increased by 39% in total and by 15% per sample. The improved accuracy of species identification was confirmed by comparing different sequences obtained from the same species. The newly registered sequences had higher similarity values (mean >0.003) than the pre-existing sequences based on both rRNA genes. These OTUs were identified at the species level based on sequences not only present in the Sea of Okhotsk but also in other areas.

**Discussion**. The results of the registration of new taxonomically verified sequences and the subsequent comparison of databases based on metabarcoding data of natural zooplankton samples clearly showed an increase in accuracy in species identification. Continuous registration of sequence data covering various environmental conditions is necessary for further improvement of metabarcoding analysis of zooplankton for monitoring marine ecosystems.

**Subjects** Ecology, Marine Biology, Molecular Biology, Zoology, Biological Oceanography
**Keywords** Japan, Marine biology, Metabarcoding, Pacific Ocean, Plankton, Species identification

## INTRODUCTION

Zooplankton are diverse and dominant organisms with high abundance and biomass in the oceans, linking primary producers to higher trophic levels including commercially important fishes (*Ward et al., 2012*). Zooplankton plays an important role in biogeochemical cycles through their behaviors such as grazing, respiration, excretion, and vertical migration (*Steinberg et al., 2008*). Approximately 7,000 species belonging to 15 phyla have been described in marine metazoan zooplankton (*Boltovskoy, Correa & Boltovskoy, 2002*), and copepods are particularly dominant and diverse with >2,700 described species (*Razouls et al., 2020*). Zooplankton has been conventionally identified based on morphological characteristics; however, species identification of zooplankton needs sophisticated expertise and technique, which is labor-intensive and time-consuming. In addition to the immature stages of zooplankton, there are still many additional new

and cryptic species in marine zooplankton, making morphological classification difficult (*Bucklin et al., 2010b*).

Molecular techniques can identify zooplankton species without depending on morphological characteristics, which enables an understanding of the community structure of zooplankton with high taxonomic resolution (*Fuller et al., 2001*; *Machida et al., 2009*). A metabarcoding approach using massively parallel next-generation sequencing (NGS) platforms is an especially promising method, which can rapidly reveal large-scale patterns of community structure in zooplankton based on massive sequence data (*Lindeque et al., 2013*; *Hirai, Tachibana & Tsuda, 2020*). Metabarcoding is also a powerful method to recover taxonomic information on damaged organisms with no clear morphological characteristics, for example, this technique has been used for diet analysis of zooplanktivorous fish (*e.g., Hirai et al., 2017a*). Metabarcoding analysis of zooplankton is thus useful to monitor marine ecosystems and understand food webs; however, a reference library is required for taxonomic classifications. According to *Lindeque et al. (2013)*, the most valuable reference library consists of well-populated sequences based on correctly identified specimens. Continuous registration of sequence data is thus indispensable for the accurate identification of thousands of zooplankton species by molecular methods.

The mitochondrial cytochrome *c* oxidase subunit I (COI) gene is a common genetic marker for molecular analysis of marine zooplankton (*Bucklin, Steinke & Blanco-Bercial, 2011*). However, the COI gene with high evolutionary rates is not suitable for designing universal primer pairs to cover a broad range of species in zooplankton (*Machida & Knowlton, 2012*; *Hirai, Shimode & Tsuda, 2013*; *Clarke et al., 2017*). The COI gene is commonly used for metabarcoding together with other molecular markers such as nuclear small and large subunit rRNA genes (*Clarke et al., 2017*; *Carroll et al., 2019*; *Berry et al., 2019*). Although 18S and 28S rRNA genes are conserved and used for resolving deep phylogenetic relationships (*Kiesling et al., 2002*; *Sonnenberg, Nolte & Tautz, 2007*; *Raupach et al., 2010*; *Bucklin, Steinke & Blanco-Bercial, 2011*; *Blanco-Bercial, Bradford-Grieve & Bucklin, 2011*), these genes are common for metabarcoding analyses due to existences of both variable and conserved regions (*Bucklin et al., 2016*). The 18S rRNA gene is the most commonly targeted for metabarcoding analysis in eukaryotes (*De Vargas et al., 2015*), and the number of 18S rRNA gene sequences is larger than that of 28S rRNA gene sequences in the public databases (*Yilmaz et al., 2014*; SILVA https://www.arb-silva.de/). The 28S rRNA gene is more variable than the 18S rRNA gene, and it is used for species identification in metazoans (*Kiesling et al., 2002*; *Sonnenberg, Nolte & Tautz, 2007*; *Raupach et al., 2010*; *Bucklin, Steinke & Blanco-Bercial, 2011*; *Blanco-Bercial, Bradford-Grieve & Bucklin, 2011*), as well as for metabarcoding of zooplankton (*Harvey et al., 2018*; *Hirai, Tachibana & Tsuda, 2020*).

Sequence differences within zooplankton species are commonly observed among different geographical regions even in the open ocean without physical barriers (*Goetze, 2003*), and a reference library specific to geographic regions of interest improves the accuracy of taxonomic identification by metabarcoding (*Questel et al., 2021*). The western North Pacific around Japan is adjacent to several water masses, and a high species diversity of zooplankton has been reported (*Tittensor et al., 2010*). In the previous study, 28S rRNA

gene sequences have been registered for >100 species of copepods in the Kuroshio area off Japan (*Hirai et al., 2015*). However, an improved database with the 18S and 28S rRNA gene sequences is also needed for other areas in Japanese waters, *e.g.*, the western North Pacific and several marginal seas such as the Sea of Okhotsk, the Sea of Japan and the East China Sea (Fig. 1). To cover the potential changes in biodiversity, it would also be beneficial to obtain sequences in different seasons. For example, based on weekly-collected zooplankton samples from the port of Mombetsu at the Sea of Okhotsk in Hokkaido (northern part of Japan) analyzed using the 18S marker by metabarcoding, unclassified sequences were commonly observed throughout the year (*Hirai et al., 2017b*). This was explained by seasonal changes in zooplankton community composition, which is strongly influenced by the transport by ocean currents such as the Kuroshio and Oyashio.

This study investigated how the accuracy of species identification by metabarcoding improves by adding taxonomically verified sequences of target rRNA genes of zooplankton from six sea areas around Japan to the public database. This was tested by using natural zooplankton samples. The effects of seasonality and ocean currents as well as the importance of including sequence data from different sea areas for improving the accuracy of zooplankton species identification is also discussed.

## MATERIALS & METHODS

### Zooplankton samples for obtaining taxonomically verified 18S and 28S rRNA gene sequences

Zooplankton samples were collected at 32 sampling stations in six areas surrounding Japan in 2018 (Fig. 1; Table 1; Table S1). These areas included the Sea of Okhotsk, the Sea of Japan, the Cold Temperate Western North Pacific (CTWNP), the Warm Temperate Western North Pacific (WTWNP), the East China Sea, and the Nansei Islands (Table 1). The bulk zooplankton samples were collected by a North Pacific Standard Plankton (NORPAC) net with 100 μm mesh, to recover epipelagic samples by vertical tows from 150 m. When the water depth was <150 m, sampling was carried out from the seafloor to the surface (Table 1). Zooplankton was preserved in 99.5% ethanol immediately for genetic analysis. The ethanol was replaced after 24 h of initial preservation, and samples were kept at −20 °C. In each sampling area, zooplankton communities were analyzed based on morphological characteristics under light microscopy by the taxonomic experts, and we selected a total of 20 to 30 numerically dominant species in each sampling area for updating the public sequence database (Table 1; Table S1). The taxonomic category (kingdom, phylum, class, order, family, genus, and species) of each zooplankton species followed the taxonomy in WoRMS (*WoRMS Editorial Board, 2021*).

### Primer selection for targeting the 18S and 28S rRNA genes

The pair of universal primers to amplify the V7–9 hypervariable regions (approximately 500 bp) was used for the 18S rRNA gene (*Sildever et al., 2019*): 18S-V7F (5′-TGGAGYGATHTGTCTGGTTDATTCCG-3′) and 18S-V9R(5′-TCACCTACGGAWACCT TGTTACG-3′). On the other hand, we designed a new forward primer for the 28S rRNA gene. For the primer design, sequence identification number information (GI)

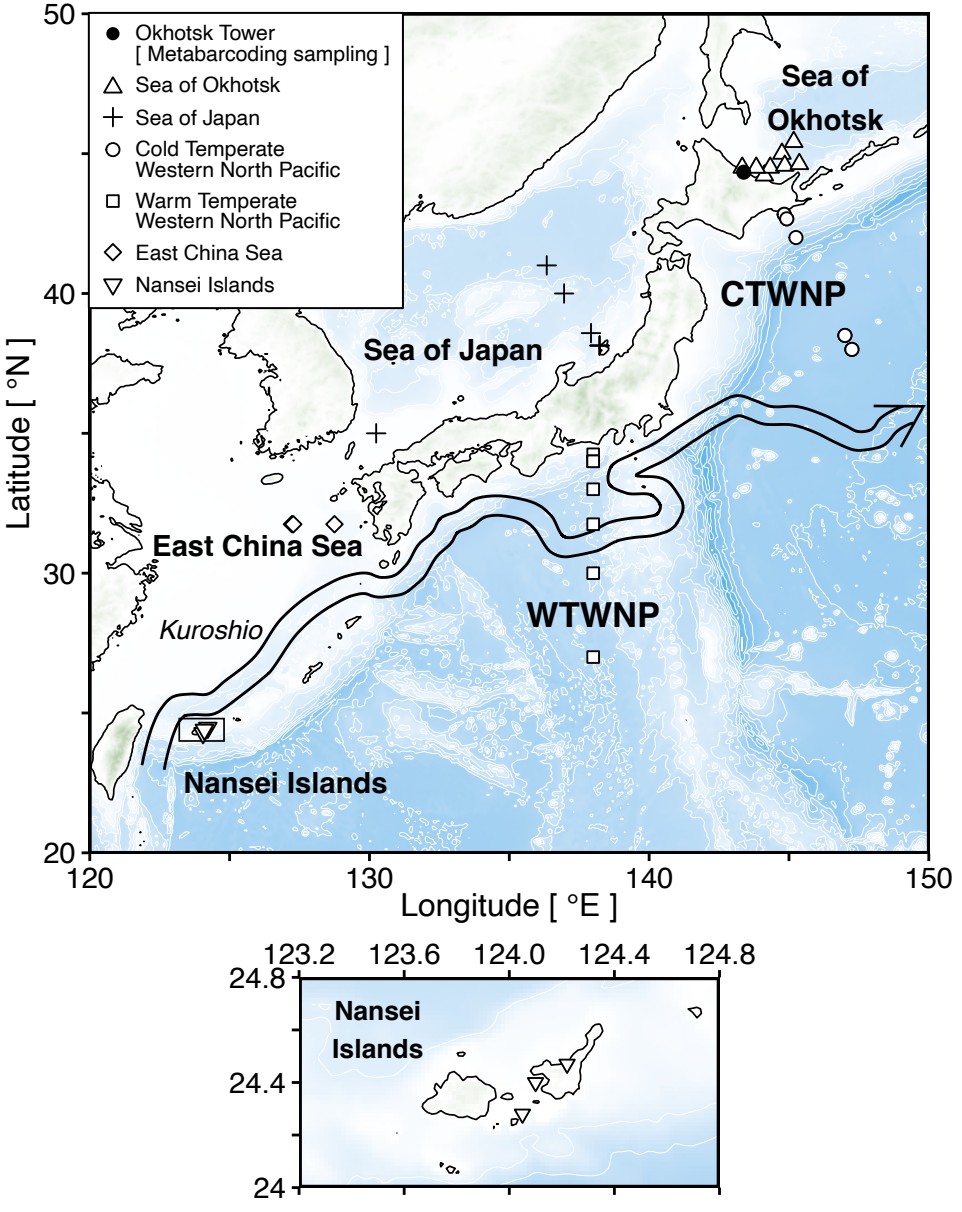

**Figure 1** **Sampling map of the stations in six sea areas surrounding Japan.** The square shows enlarged views of research stations in the Nansei Islands. Triangle (△), plus (+), circle (○), square (□), diamond (◇), and inverted triangle (▽) indicate six sea areas, *i.e.,* the Sea of Okhotsk, the Sea of Japan, the Cold Temperate Western North Pacific (CTWNP), the Warm Temperate Western North Pacific (WTWNP), the East China Sea, and the Nansei Islands, respectively. Black circle (●) shows the Okhotsk Tower which is the metabarcoding sampling station. The coordinates of all stations are available in Table 1. The large arrow denotes the Kuroshio current path in May 2018 based on the Quick Bulletin of Ocean Conditions issued by the Japan Coast Guard (https://www1.kaiho.mlit.go.jp/KANKYO/KAIYO/qboc/index_E.html).

of Amphipoda, Chaetognatha, Cnidaria, Copepoda, Decapoda, Euphausiacea, Mollusca, Mysidacea and Ostracoda for 28S rRNA gene was downloaded from the National Center for Biotechnology Information nucleotide database (NCBI nt: https://www.ncbi.nlm.nih.gov/)

**Table 1** **Sampling information for the locations of taxonomically verified sequences and metabarcoding of zooplankton.** Sample indicates which samples were used for analysis. Sea area shows the sampling area (and abbreviations) for this study. Depth shows the depth of the sampling station, not the net towing depth for zooplankton collecting.

| Sample | Sea area | Cruise or location | Station | Longitude [°E] | Latitude [°N] | Depth [m] | Date | SST [°C] |
|---|---|---|---|---|---|---|---|---|
| Individual zooplankton | Sea of Okhotsk | HK1805 or WK1809 | N1 | 144.12 | 44.22 | −55 | May and Sep 2018 | 8.5, 16.7 |
| | | | N2 | 144.33 | 44.50 | −734 | May and Sep 2018 | 6.7, 12.2 |
| | | | N3 | 144.75 | 45.00 | −1,553 | May and Sep 2018 | 7.2, 13.2 |
| | | | N4 | 145.17 | 45.42 | −2,977 | May and Sep 2018 | 6.3 |
| | | | S1 | 143.33 | 44.50 | −57 | May and Sep 2018 | 9.1, 17.4 |
| | | | S2 | 143.83 | 44.50 | −164 | May and Sep 2018 | 8.1, 15.2 |
| | | | S3 | 144.85 | 44.57 | −1,655 | May and Sep 2018 | 6.3, 13.4 |
| | | | S4 | 145.37 | 44.63 | −2,239 | May and Sep 2018 | 8.4 |
| | Sea of Japan | SHU1804 | SI01 | 138.23 | 38.14 | −130 | 18 Apr 2018 | 10.7 |
| | | | SI07 | 136.96 | 40.00 | −2,060 | 17 Apr 2018 | 7.7 |
| | | | SI09 | 136.34 | 41.00 | −3,310 | 17 Apr 2018 | 4.5 |
| | | TY1806 | SI01 | 138.23 | 38.14 | −130 | 13 Jun 2018 | 17.4 |
| | | TY1809 | SI01 | 138.23 | 38.14 | −130 | 21 Sep 2018 | 23.7 |
| | | | SI03 | 137.92 | 38.58 | −1,890 | 20 Sep 2018 | 23.4 |
| | | YK1801 | St.89 | 130.24 | 35.00 | −125 | 30 Apr 2018 | 16.1 |
| | Cold Temperate Western North Pacific (CTWNP) | WK1805 | A01 | 144.83 | 42.83 | −94 | 18 May 2018 | 5.0 |
| | | | A02 | 144.92 | 42.67 | −764 | 18 May 2018 | 6.1 |
| | | | A19 | 147.00 | 38.50 | −5,438 | 13 May 2018 | 15.4 |
| | | WK1807 | A01 | 144.83 | 42.83 | −94 | 18 Jul 2018 | 14.3 |
| | | | A02 | 144.92 | 42.67 | −764 | 18 Jul 2018 | 15.3 |
| | | | A05 | 145.25 | 42.00 | −4,382 | 17 Jul 2018 | 15.4 |
| | | | A21 | 147.25 | 38.00 | −5,559 | 13 Jul 2018 | 21.4 |
| | | HK1809 | A19 | 147.00 | 38.50 | −5,438 | 3 Oct 2018 | 20.8 |
| | | | A21 | 147.25 | 38.00 | −5,559 | 2 Oct 2018 | 22.6 |
| | Warm Temperate Western North Pacific (WTWNP) | SY1804 | C2700 | 138.00 | 27.00 | −4,864 | 23 Apr 2018 | 23.0 |
| | | | C3300 | 138.00 | 33.00 | −4,145 | 19 Apr 2018 | 18.1 |
| | | | C3415 | 138.00 | 34.25 | −824 | 18 Apr 2018 | 17.4 |
| | | SY1811 | C3000D | 138.00 | 30.00 | −4,056 | 12 Nov 2018 | 26.0 |
| | | | C3145D | 138.00 | 31.75 | −3,781 | 10 Nov 2018 | 25.2 |
| | | | C3400N | 138.00 | 34.00 | −1,501 | 15 Nov 2018 | 21.5 |
| | | | C3400D | 138.00 | 34.00 | −1,501 | 16 Nov 2018 | 21.5 |
| | East China Sea | YK1801 | St.74 | 127.23 | 31.75 | −124 | 27 Apr 2018 | 18.5 |
| | | YK1805 | St.24 | 127.28 | 31.75 | −128 | 30 Jun 2018 | 24.3 |
| | | YK1806 | St.9 | 128.75 | 31.75 | −767 | 20 Jul 2018 | 29.6 |
| | Nansei Islands | | U5 | 124.22 | 24.47 | −25 | 23 May, 25 Jun, 24 Jul, 21 Aug, 2 Oct, 6 Nov, 6 Dec 2018 | 24.5–29.6 |

| Sample | Sea area | Cruise or location | Station | Longitude [°E] | Latitude [°N] | Depth [m] | Date | SST [°C] |
|---|---|---|---|---|---|---|---|---|
| | | | S11 | 124.05 | 24.28 | −36 | 24 May, 8 Aug, | 25.4–30.3 |
| | | | N1 | 124.10 | 24.40 | −55 | 15 Nov 2018 | 25.0–30.4 |
| Metabarcoding | Sea of Okhotsk[*] | Mombetsu | Okhotsk Tower | 143.38 | 44.34 | −10 | Apr 2012–Mar 2016 | −1.7–22.2 |

**Notes.**
SST, Sea Surface Temperature.
*See Table S2 for details of the sampling data.

($n = 46,217$). The GI data was converted to fasta format using blastdbcmd (version 2.6.0) based on the NCBI nt downloaded on August 20, 2020. Overlapping GI-s were removed ($n = 46,210$). The sequences containing D1-2 regions were extracted using a primer pair of F63 (5′-GCATATCAATAAGCGGAGGAAAAG-3′) and R635 (5′-GGTCCGTGTTTCAAGACGG-3′) (*Kiesling et al., 2002*) by *in silico* PCR with the aid of ecoPCR v0.8.0 (*Bellemain et al., 2010*). In this analysis, three bp mismatches in each primer sequence were allowed, but no mismatch was allowed in the three bases at the 3′-end. Sequences that contained poly-N longer than 4 bp in the amplified region were excluded from the data ($n = 1,339$). The alignment was done by MAFFT (version: 7.402-with-extensions) (*Katoh & Standley, 2013*). The nucleotide homology was checked by GENETYX version 15 (Genetyx, Tokyo, Japan) to find conservative regions to design a new forward primer. The forward primer of Zoop_28S_D2F (5′-GAGAGTTCAAVAGTACGTGAA-3′) was determined by considering the sequence variability and the length of the amplicon (<500 bp), in combination with the reverse primer of R635 to amplify the D2 region in 28S rRNA gene (*Kiesling et al., 2002*).

## Sequencing and registration of the 18S and 28S rRNA genes

Genomic DNAs were extracted from an individual of morphologically identified species (Table S1) using the QuickGene-810 (Fujifilm, Tokyo, Japan) according to the manufacturer's protocol. PCR amplification was carried out on a thermal cycler (PC-808, ASTEC, Fukuoka, Japan) with a reaction mixture consisting of 1 μL template DNA, 1 μM each of 18S (18S-V7F and 18S-V9R) and 28S (Zoop_28S_D2F and R635) rDNA primer sets, 0.2 mM of each dNTP, 1 × PCR buffer, 1.5 mM $Mg^{2+}$, 1U KOD-Plus-Ver.2 (TOYOBO, Osaka, Japan), and DNA/RNA-free $dH_2O$ bringing up the volume to 25 μL. The PCR for both markers was carried out separately. The PCR cycling conditions for the 18S marker were as follows: 2 min at 94 °C, 30 cycles at 94 °C for 15 s, 56 °C for 30 s, and 68 °C for 40 s. The same PCR conditions were used for amplifying the 28S marker, except for the annealing temperature (58 °C). Sequences of the target regions were obtained by the direct Sanger sequencing method using the Dynamic ET terminator cycle sequencing kit (GE Healthcare, Little Chalfont, UK) and a DNA sequencer (ABI3730; Applied Biosystems, Waltham, MA, USA).

When sequences were not successfully obtained by the direct sequencing method, we used the sub-cloning method (Table S1). Because KOD-Plus-Ver.2 is a thermostable polymerase containing extensive 3′ to 5′ exonuclease activity and results in PCR products with a blunt end, the amplicons were further treated for adenine addition to the 3′ end per A-tailing

procedure following the pGEM-T® Easy Vector System technical manual. *Ajani et al. (2022)* previously described the operation. Specifically, these 3′-adenine overhang products were immediately ligated to the pGEM-T® Easy Vector (Promega, Madison, WI, USA) and transformed into DH5$\alpha$ cells (Promega, Madison, WI, USA) following the manufacturer's protocol. After incubating the plates at 37 °C for colony growth, six white colonies were randomly chosen from each plate. The insertion of the target region was confirmed by colony PCR, and three positive PCR products in each sample were processed for the Sanger sequencing using the universal primers: U19 (F): (5′-GGTTTTCCCAGTCACGACG-3′) and M13 (R) (5′-CAGGAAACA GCTATGAC-3′; *Vassart et al., 1987*) .

The sequences were aligned using MEGA version 10 (*Kumar et al., 2018*) and the consensus sequences were obtained for each species. The BLAST search was performed to confirm the availability of sequences of the same species on the GenBank. All newly obtained sequences were deposited into the DDBJ databank (accession numbers: LC581890–LC582220).

### Sampling for metabarcoding analysis based on field samples

The metabarcoding samples of zooplankton were collected weekly at the Okhotsk Tower in the Port of Mombetsu (Fig. 1) facing the northeastern coastal Sea of Okhotsk. Zooplankton samples were collected from the bottom (10 m depth) to the surface using NORPAC net with 335 $\mu$m mesh size, preserved at −20 °C until the following experiments. We used a total of 118 samples for obtaining the 18S rRNA gene sequences from 10 April 2012 to 19 February 2016 and a total of 194 samples for obtaining the 28S rRNA gene sequences from 10 April 2012 to 16 March 2016 (Fig. 1; Table 1; Table S2).

### DNA extraction, library preparation, and sequencing for metabarcoding

Genomic DNAs were extracted from bulk frozen samples using the QuickGene-810 (Fujifilm, Tokyo, Japan) according to the manufacturer's protocol. Samples were split for DNA extraction in the case of a large number of specimens. For metabarcoding analysis using the MiSeq 300PE platform (Illumina, San Diego, CA, USA), we followed the workflow of "16S metagenomic sequencing library preparation: preparing 16S ribosomal gene amplicons for the Illumina MiSeq system" distributed by Illumina (part no. 15044223 Rev. B). A two-step PCR approach was used to construct the paired-end libraries of PCR amplicons, which were flanked by primer-binding sites for sequencing, dual-index (*i.e.*, barcode) sequences, and adapter sequences for binding to the flow cells of the MiSeq platform.

The first PCR amplified the target regions using the following primer pairs: 5′-ACACTCTTTCCCTACACGACGCTCTTCCGATCT-3′ + 18S or 28S marker (forward) and 5′-GTGACTGGAGTTCAGACGTGTGCTCTTCCGA TCT-3′ + 18S or 28S marker (reverse). We used the same primer sequences as for obtaining the taxonomically verified sequences for 18S (18S-V7F and 18S-V9R) and 28S (Zoop_28S_D2F and R635) rRNA genes. The first PCR was performed using a thermal cycler (PC-808; ASTEC, Fukuoka, Japan) in a 25-$\mu$L reaction mixture containing 1.0 $\mu$L template DNA (<1 ng), 0.2 mM

of each dNTP, $1 \times$ PCR buffer, 1.5 mM $Mg^{2+}$, 1.0 U KOD-Plus-ver.2 (Toyobo, Osaka, Japan), and 1.0 µM of each primer. The PCR cycling conditions for the 18S marker were as follows: initial denaturation at 94 °C for 3 min, followed by 30 cycles at 94 °C for 15 s, 56 °C for 30 s, and 68 °C for 40 s. PCR amplification was verified by 1.5% agarose gel electrophoresis. The same PCR conditions were used for amplifying the 28S marker, except for the annealing temperature (58 °C). The PCR products were purified using an Agencourt AMPure XP (Beckman Coulter, Brea, CA, USA) and eluted in 25 µL TE buffer according to the manufacturer's protocol.

After purification, the PCR products were diluted 1:5 in Milli-Q water and used as a template for the second PCR. The following primers: 5′-AATGATACGGCGACCACCGAGATC TACAC-8 bp index-ACACTCTTTCCCTACACGACGC-3′ (forward) and 5′-CAAGCA GAAGACGGCATACGAGAT-8 bp index-GTGACTGGAGTTCAGACGTGTG-3′ (reverse) were used in the second PCR. The 8-bp segments represent dual-index sequences for recognizing each sample, the 5′ end-sequences are adapters that allow the final product to bind or hybridize to short oligonucleotides on the surface of the Illumina flow cell, and the 3′ end-sequences are priming sites for MiSeq sequencing. The second PCR was performed in the reaction mixture volume of 50 µL including 2.0 µL of purified and diluted PCR product. The PCR cycling conditions were as follows: initial denaturation at 94 °C for 3 min, followed by 10–12 cycles at 94 °C for 15 s, 59 °C for 30 s, and 68 °C for 40 s. PCR amplification was again verified by agarose gel electrophoresis, and the PCR products were purified using an Agencourt AMPure XP (Beckman Coulter, Brea, CA, USA). The amplified PCR products were quantified by Qubit (Invitrogen, USA), and the indexed PCR products from the second PCR were pooled in equal concentrations and stored at −30 °C until used for sequencing. Sequencing runs were carried out using MiSeq Reagent Kit v3 on the Illumina MiSeq platform.

## Bioinformatics analyses and operational taxonomic unit picking

Nucleotide sequences were demultiplexed depending on the 5′-multiplex identifier (MID) tag and primer sequences using the default format in MiSeq. Trimming was performed as previously described in *Nagai et al. (2022)*. Specifically, the sequences containing palindrome clips longer than 30 bp and homopolymer longer than 9 bp were trimmed from the sequences at both ends. The 3′ tails with an average quality score of less than 30 at the end of the last 25-bp window were also trimmed from each sequence. The 5′ and 3′ tails with an average quality score of less than 20 at the end of the last window were also trimmed from each sequence. Sequences longer than 300 bp were truncated to 300 bp by trimming the 3′ tails. The trimmed sequences shorter than 250 bp were filtered out. The demultiplexing and trimming were performed using Trimmomatic version 0.35 (http://www.usadellab.org/cms/?page=trimmomatic). The remaining sequences were merged into paired reads using Usearch version (http://www.drive5.com/usearch/). The clustering of merged sequences was performed using the cluster-features-de-novo method included in the vsearch plugin of Qiime2 version 2020.8 (https://docs.qiime2.org/2022.11/plugins/available/). The Identity option was specified at 0.990. Erroneous and chimeric sequences were detected and removed using

the cluster-features-de-novo method included in the vsearch plugin of Qiime2 version 2020.8 (https://docs.qiime2.org/2022.11/plugins/available/). An OTU-count table was made using the feature-table and filter-features methods in Qiime2 with an option of removal of singleton by "–p-m-frequency 2". Demultiplexed and filtered but untrimmed sequence data were deposited into the DDBJ Sequence Read Archive(access no. DRA010320).

## Taxonomic identification of the OTUs

A subset of nucleotide databases was selected from the NCBI database based on the following conditions for taxonomic identifications of OTUs using a BLAST search. The taxonomic information containing "ribosomal", "rrna", and "rdna", were extracted, but those containing "protein", "metagenome", "uncultured" and "environmental" was excluded. The sequences of retrieved GenBank IDs from the Nucleotide database downloaded from the NCBI FTP server were extracted on 13 June 2022 and used to construct a template sequence database. The taxonomic identification of each OTU was subsequently performed by BLAST search (*Cheung et al., 2010*) with NCBI BLAST+ 2.10.1+ (*Camacho et al., 2009*) using the default parameters. The nucleotide subset described above was used for the reference database and all OTU-representative sequences as the query. The taxonomic information shown as tophit_name was obtained from the BLAST hit with the top bitscores for each query sequence, and the taxonomic information showing the same top hit similarity were then merged. Namely, when an OTU had the same bitscore and the same top hit similarity, the top hit names were enumerated accordingly. The removal of sequences containing errors was imperfect after the successive processes of metabarcoding data treatment; sequences containing different types of errors derived from the original ones remained in the following analytical steps. Therefore, we recognized these sequences as artificially formed OTUs showing the same top hit name but slightly different sequences, resulting in showing the different top hit identities, as demonstrated in the result of pyrosequencing using only *Escherichia coli* MG1655 as a reference template (*Kunin et al., 2010*). To avoid overestimation of the OTUs, these artificial OTUs were merged into a single OTU with the greatest similarity score. Species were identified based on the criterion that only when the top hit similarity was >0.98 for 18S and 28S rRNA gene sequences in Blast search and the OTU showed a single top hit with a full species name. When a single species was clustered together with other records identified as sp. from the same genus in an OTU (multi-top hit), these OTUs were regarded as the OTUs identified at species levels.

## Comparison of improvement in species identification between reference databases with and without new sequences

The improvement in the detection power of zooplankton species was investigated based on the metabarcoding data from field samples from the Sea of Okhotsk (Okhotsk Tower) (Fig. 1; Table 1; Table S2). Two reference databases were used for confirming the improved accuracy of species identification. The conventional (Old) database was the above template sequence database from GenBank, which did not include newly registered sequences. The new database (New) contained the taxonomically verified sequences and the template database. BLAST searches were performed on the same metabarcoding sequences using

both reference databases with the BLAST search settings as described above (the top hit similarity >0.98) for both target genes. To verify the intraspecific variation between the sequences, the OTUs were not grouped into species. To confirm the improved accuracy of species identification, the number of OTUs detected per sample was tested for a statistically significant difference between the old and new databases by the Wilcoxon rank sum test. Furthermore, to clarify the difference in detection power for each sample, the number of OTUs detected in the new database minus the number of OTUs detected in the old database (delta OTUs) was calculated: a positive value indicates improved accuracy, a negative number indicates no improvement and zero indicates no change. We also tested for a statistically significant difference by the Wilcoxon test in the number of species detected between seasons using the Delta OTUs. Statistical analyses were conducted using R (*R Core Team, 2022*).

## RESULTS

### Dominant zooplankton species used for obtaining taxonomically verified sequences

In total, 96 species, including two unidentified species (represented by four individuals and four sequences), were selected as dominant zooplankton species from six areas for obtaining taxonomically verified sequences of 18S and 28S rRNA genes (Fig. 2). From the dominant species, 94 belonged to Arthropoda and 2 to Chaetognatha. In Arthropoda, 91 species belonged to Copepoda, followed by Amphipoda (2 species), Euphausiacea (2 species), and Onychopoda (1 species) (Fig. 2). Copepoda species belonged to the orders of Calanoida(16 families and 31 genera), Cyclopoida (3 families and 7 genera), and Harpacticoida (3 families and 3 genera) (Fig. 2). More than half (57%) of those were registered from a single sea area, whereas 25% were registered from two sea areas (Table S1). At the same time, 18% were registered from three or more sea areas. Four species were also registered from five sea areas: *Calanus sinicus*, *Microsetella norvegica*, *Oithona similis*, and *Oncaea venusta* (Table S1). Detailed information on species and sampling locations for taxonomically verified 18S and 28S rRNA gene sequences is available in Table S1.

### Registration of the taxonomically verified sequences for the target genes

A total of 166 18S rRNA gene sequences were obtained from all 96 species, including four sequences from two unidentified species associated with *Paracalanus* sp. from the WTWNP, CTWNP, and the Nansei Islands and *Metridia* sp. from the Sea of Okhotsk (Table S1). From the 162 sequences belonging to 94 morphologically identified species, 30% were already registered in the public database, *i.e.,* the sequences matched with a single species by BLAST (“Pre-exist” of Total in Fig. 3, Table 2). The remaining 70% of the sequences (associated with 66 species from 38 genera) were not available in the public database (“New” of Total in Fig. 3, Table 2). Of these, 62% had a match at genus level(“New: match in genus” of Total in Fig. 3, Table 2), and 8% had a match with the higher rank (family or more) (“New: match in higher rank” of Total in Fig. 3, Table 2). In addition, 39 species had multiple sequences from different individuals and 36 of those were

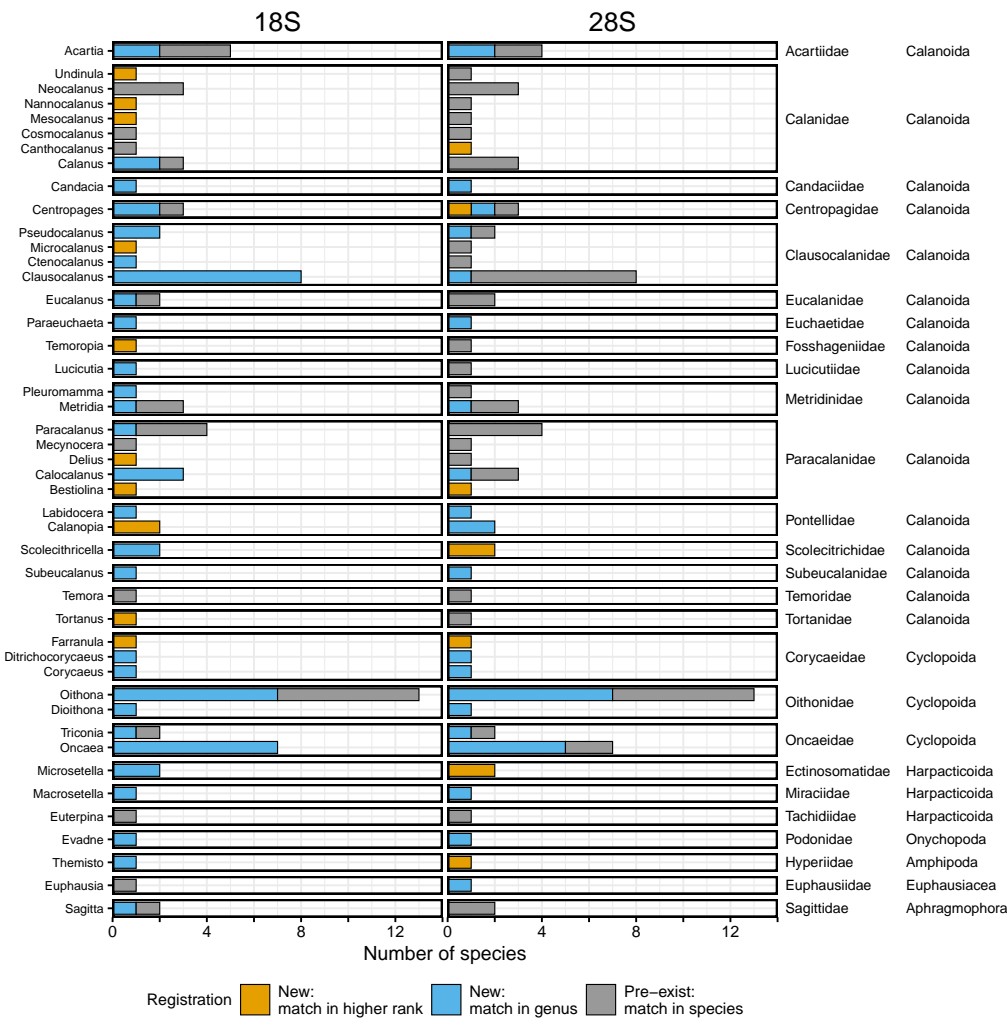

**Figure 2  Number of morphologically verified species per genus in registered zooplankton sequences.** The number of morphologically identified species per genus based on the 18S (left, 166 sequences from 96 species) and the 28S (right, 165 sequences from 95 species) rRNA genes registered in this study. The family and order (right side) match with the genus (left side). Colors indicate that the sequences matched with a higher rank (family or more, orange), genus (blue), and species (*i.e.,* pre-exist sequences, grey) in the public database using BLAST.

common sequences within a species (except for *Mesocalanus tenuicornis*, *Oithona nana*, and *Oncaea venusta*).

Based on the 18S rRNA genes, the newly registered sequences changed regionally from 12 to 25 in each area (Fig. 3, Table 2). The number of species with unregistered sequences in the public database was higher in the southern waters, including the WTWNP (83%), the East China Sea (79%), the Sea of Japan (74%), and the Nansei Islands (69%). Relatively low proportions of species with unregistered sequences were observed in the cold waters of the Sea of Okhotsk (50%) and the CTWNT (62%).

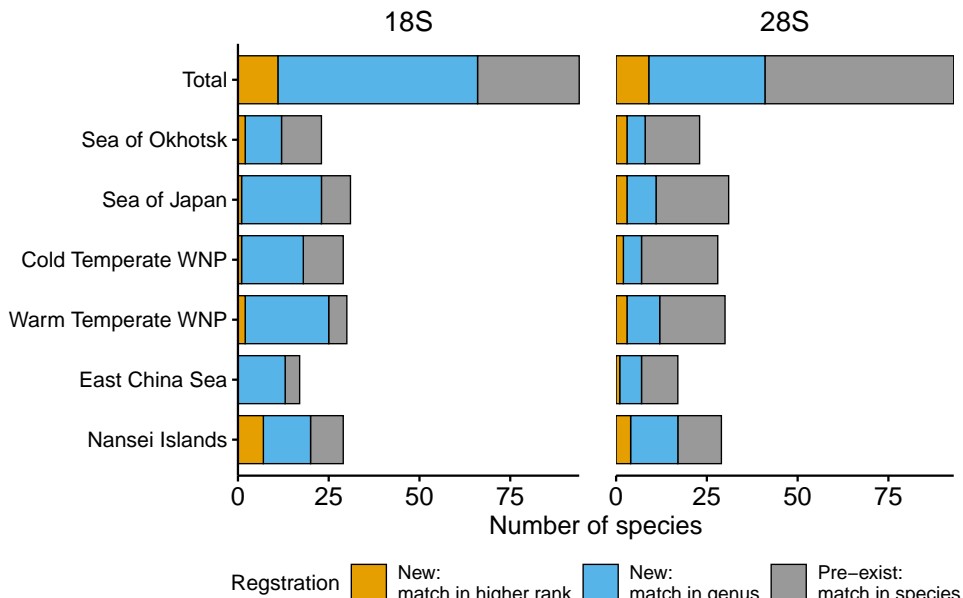

**Figure 3 Number of morphologically verified zooplankton species from six sea areas for which sequences were registered.** 18S (left) and 28S (right) rRNA gene sequences from all sampling stations (Total) and six sea areas. Colors indicate that the sequences match with a higher rank (family or more, orange), genus (blue), and species (*i.e.*, pre-exist sequences, grey) in the public database using BLAST.

A large number of newly registered species were observed in the genus *Oithona* (for seven out of 13 species), *Oncaea* (for seven out of seven species), and *Clausocalanus* (for eight out of eight species) in Copepoda (Fig. 2). These three genera occupied 56% and 69% of newly registered species in the WTWNP and the East China Sea, respectively (Table S1). In addition to *Clausocalanus* species, one species in the genera *Ctenocalanus* ("New: match in genus") and *Microcalanus* ("New: match in higher rank"), and two species in the genus *Pseudocalanus* ("New: match in genus") in the Clausocalanidae were also previously unregistered (Fig. 2). A relatively large number of newly registered species were observed in the family Paracalanidae (six species) and Corycaeidae (three species) (Fig. 2).

For the 28S rRNA gene, a total of 165 taxonomically verified sequences of 95 were registered in the public database (Table S1). The sequence of one species (*Acartia omorii* from the CTWNP) was not obtained successfully. Therefore, 161 sequences belonging to 93 species, except for two unidentified species associated with *Paracalanus* sp. from the WTWNP, CTWNP, and the Nansei Islands and with *Metridia* sp. from the Sea of Okhotsk with four sequences, were used for further analysis (Table S1). A total of 60% of the sequences associated with a single species were already in the public database, *i.e.*, the sequences matched in species by BLAST ("Pre-exist" of Total in Fig. 3, Table 2). Further 40% of the sequences associated with a single species were not available in the public database ("New" in Fig. 3, Table 2). Of those, 30% had a match at the genus level ("New: match in genus" of Total in Fig. 3, Table 2), and 10% had a match with the

**Table 2 Summary of the taxonomically verified and newly registered zooplankton sequences from this study.** 'Sequence' indicates the number of sequences, and 'Species' indicates the number of groups by species which includes duplicate sequences of the same species from other specimens. The table denoted our registered sequence was classified to what taxonomic level on the public database by BLAST. 'Pre-exist' indicates a sequence with a complete match at a species level based on the public database, and 'New' a match with taxonomic criteria of genus or higher level (*i.e.*, the sequences for which the species did not exist in the public database) database. Abbreviations: CTWNP, Cold Temperate Western North Pacific; WTWNP, Warm Temperate Western North Pacific.

| rDNA | Sea area | Species | | | | Sequences | | | |
|---|---|---|---|---|---|---|---|---|---|
| | | Total | Pre-exist | New | | Total | Pre-exist | New | |
| | | | | genus | higher rank | | | genus | higher rank |
| 18S | Total | 94 | 28 | 55 | 11 | 162 | 49 | 100 | 13 |
| | Sea of Okhotsk | 23 | 11 | 10 | 2 | 24 | 12 | 10 | 2 |
| | Sea of Japan | 31 | 8 | 22 | 1 | 31 | 8 | 22 | 1 |
| | CTWNP | 29 | 11 | 17 | 1 | 29 | 11 | 17 | 1 |
| | WTWNP | 30 | 5 | 23 | 2 | 30 | 5 | 23 | 2 |
| | East China Sea | 18 | 4 | 14 | 0 | 19 | 4 | 15 | 0 |
| | Nansei Islands | 29 | 9 | 13 | 7 | 29 | 9 | 13 | 7 |
| 28S | Total | 93 | 52 | 32 | 9 | 161 | 97 | 48 | 16 |
| | Sea of Okhotsk | 23 | 15 | 5 | 3 | 24 | 16 | 5 | 3 |
| | Sea of Japan | 31 | 20 | 8 | 3 | 31 | 20 | 8 | 3 |
| | CTWNP | 28 | 21 | 5 | 2 | 28 | 21 | 5 | 2 |
| | WTWNP | 30 | 18 | 9 | 3 | 30 | 18 | 9 | 3 |
| | East China Sea | 18 | 10 | 7 | 1 | 19 | 10 | 8 | 1 |
| | Nansei Islands | 29 | 12 | 13 | 4 | 29 | 12 | 13 | 4 |

higher rank (family or more) ("New: match in higher rank" of the total in Fig. 3, Table 2). Furthermore, 39 species had multiple sequences from different individuals, and 34 of those were common sequences within a species (except for *Clausocalanus parapergens*, *Macrosetella gracilis*, *Neocalanus plumchrus*, *Oncaea waldemari*, and *Themisto japonica*).

Similarly to the species registered by the 18S rRNA gene sequences, the highest proportion of newly registered sequences for 28S was associated with a single species originated from the Nansei Islands (59%), followed by the East China Sea (41%) and the WTWNP (40%) (Fig. 3, Table 2). Unlike the sequences obtained by the 18S marker, the sequences obtained from the WTWNP based on the 28S marker showed a relatively high proportion of species already on the public database (60%). The newly registered sequences for a single species were the lowest for the CTWNP (25%), whereas for the Sea of Okhotsk, 35% of the sequences were associated with a single species and for the Sea of Japan 32% of the sequences were newly registered.

The genera with the highest number of newly registered sequences associated with a single species belonged to *Oithona* (54%) and *Oncaea* (71%) from the order Cyclopoida (Fig. 2). These species were mainly collected in the southern part of Japan including the Nansei Islands (three *Oithona* and one *Oncaea* species), the East China Sea (one *Oithona* and three *Oncaea* species), and the WTWNP (three *Oithona* and five *Oncaea* species) (Table S1). These two genera occupied 66% and 57% of the newly registered sequences associated with a single species for the WTWNP and the East China Sea, respectively (Table S1). Most

of the species in the genus *Clausocalanus*, which showed high proportions of unregistered sequences based on the 18S rRNA gene, had a low proportion of unregistered sequences number (12%) based on the 28S rRNA gene (Fig. 2).

## Testing the improvement of taxonomic classification based on metabarcoding data from a field sample

In the metabarcoding analysis based on the 18S marker from the Sea of Okhotsk (Okhotsk Tower), 269 OTUs of Metazoa were detected with >0.980 as a single top-hit species based on the database containing newly registered sequences. The sequence identity by BLAST hit ranged from 0.980 to 0.990 in 79 OTUs, 0.991–0.999 in 172 OTUs, and 1.000 in 18 OTUs. Because the sequences registered for the dominant zooplankton taxa were associated with Arthropoda and Chaetognatha, we focused on the 97 OTUs associated with those phyla in the metabarcoding analysis.

Based on the 18S marker, from the OTUs detected in all 117 samples 79 were detected based on the conventional (old) reference database and 92 OTUs based on the new reference database including newly registered sequences (Table 3). 73 OTUs were common to the old and new databases, 19 OTUs were unique OTUs for the new database, and 6 OTUs for the old database (Table 3). In the case of 4 out of 6 OTUs in the old database, different OTUs of the same species were detected from the new database, and 3 OTUs were replaced by newly registered sequences with improved similarity (*Acartia negligens*, *Centropages abdominalis*, and *Ditrichocorycaeus anglicus*) (see below, Table S3). The new database improved the zooplankton species detection by 18%. Based on the old database, the number of OTUs per sample was on average 14.2 ± 8.9 with a minimum of 2 and a maximum of 50 OTUs (Fig. 4; Table 3). Based on the new database, the average number of OTUs per sample was 15.6 ± 10.2 with a minimum of and a maximum of 53 OTUs. Per sample, the new database increased species detection by 10%, however, there was no statistically significant difference between the identification success based on the old and new databases (Fig. 4). For the majority of samples, the identification success improved, when using the new database (Fig. 5, Table S2). The delta OTUs, OTUs from the new database minus the number of OTUs from the old database, was on average 1.4 ± 1.6 with a minimum of -2 and maximum of 8, which were above zero for the majority of samples with the exception of four samples (Fig. 5).

Of the 92 OTUs detected based on the new database, 19 were detected only based on the newly registered sequences (Table 3). Four species were detected as different OTUs based on the new and old databases, of which *Acartia negligens* and *Centropages abdominalis* were detected by both databases (Table S3). The newly registered sequences had higher similarity (≥0.002) among the sequences associated with a single species compared to the existing sequences. This increased the number of occurrences and detection for some species (*C. abdominalis*), whereas for some species it remained the same (*A. negligens*). For 19 species detected in metabarcoding samples from the Sea of Okhotsk (Okhotsk Tower) based on the newly registered sequences, only the sequences for five species (*Calanus pacificus*, *Corycaeus affinis*, *Eucalanus bungii*, *Pseudocalanus newmani*, *M. norvegica*) were registered from the Sea of Okhotsk (Table S3). Sequences of the 18S rRNA gene for the remaining 14 species

**Table 3 Overview of OTUs associated with a single zooplankton species detected from metabarcoding samples based on the old and new reference databases.** Zooplankton composition of metabarcoding samples from Okhotsk Tower based on the reference databases with (new) and without (old) newly registered sequences, performed separately for the 18S and 28S rRNA gene sequences. 'Total' is the number of OTUs in the old and new databases combined; 'New' and 'Old' are the number of OTUs in each database. 'Difference' is the number of OTUs in the new database minus the number of OTUs in the old database, representing the increase in the number of detections in the new database. The OTU number detected in both databases ('Shared OTUs') or only in the respective database are also shown ('Unique OTUs'). However, most OTUs that appeared in the old database were also detected at the species level in the new database (see text for details). 'OTU per sample' shows the respective OTU number (mean, standard deviation, minimum, and maximum) per sample.

| rRNA | Number of detected OTUs | | Shared OTUs | Unique OTUs | OTUs per sample | | | |
|---|---|---|---|---|---|---|---|---|
| | | | | | Mean | SD | Min | Max |
| 18S | Total | 98 | 73 | | | | | |
| | New | 92 | | 19 | 15.6 | 10.2 | 2 | 53 |
| | Old | 79 | | 6 | 14.2 | 8.9 | 2 | 50 |
| | Difference | 13 | | | 1.4 | 1.6 | −2 | 8 |
| 28S | Total | 109 | 44 | | | | | |
| | New | 89 | | 45 | 12.1 | 4.7 | 3 | 26 |
| | Old | 64 | | 20 | 10.5 | 4 | 1 | 21 |
| | Difference | 25 | | | 1.7 | 1.5 | −1 | 8 |

were registered from other areas, including 13 species associated with Copepoda and one species associated with Branchiopoda.

In total, 160 OTUs associated with Metazoa were detected based on the 28S rRNA gene metabarcoding, with >0.980 similarity to a single top-hit species in the BLAST search including newly registered sequences. The sequence identities were 0.981–0.990 in 32 OTUs, 0.991–0.999 in 98 OTUs, and 1.000 in 30 OTUs. For further analysis, 88 OTUs in the phyla Arthropoda and Chaetognatha were used.

Based on the 28S marker, 64 OTUs were detected using the old database, and 89 OTUs using the new database (Table 3). The identification improved by 42% based on the new database compared to the old database. 44 OTUs were common to the old and new databases, 45 OTUs were unique for the new database, and 20 OTUs for the old database (Table 3). For 19 out of 20 unique OTUs in the old database, different OTUs of the same species were also detected based on the new database, and all OTUs were replaced by newly registered sequences with improved similarity (see below, Table S3). For example, an OTU associated with *Ditrichocorycaeus anglicus* was only detected based on the old database, whereas, a closely related species, *Ditrichocorycaeus affinis*, was detected based on the new database (a newly registered sequence with higher similarity). Based on the old database, the average number of OTUs per sample was 10.5 ± 4.0 with a minimum of one OTU and a maximum of 21 OTUs (Fig. 4; Table 3). Based on the new database, the average number of OTUs for each sample was 12.1 ± 4.7 with a minimum of 3 OTUs and a maximum of 26 OTUs. Per sample, the improvement was 20%, when using the new database. The difference in identification success between the two databases was statistically significant ($p < 0.01$) (Fig. 4; Table 3). Three samples contained species that were detected only based on the new database. The delta OTUs was on average 1.7 ± 1.5 with a minimum of −1, and

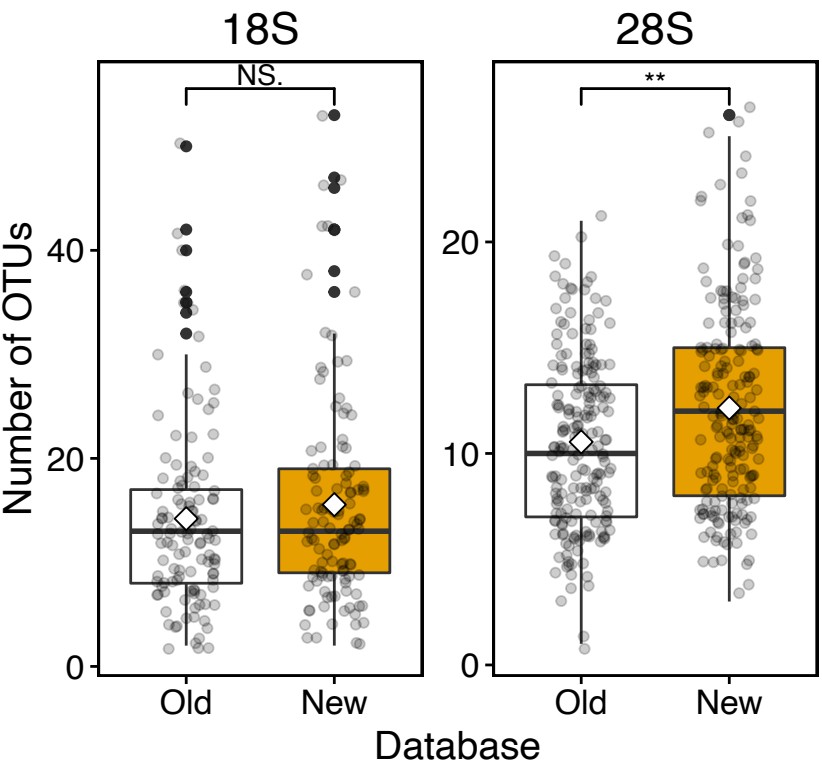

**Figure 4 Boxplots of OUT numbers associated with a single zooplankton species detected based on old and new reference databases.** Old (white) and New (orange) indicate the reference database without and with taxonomically verified and newly registered sequences, respectively. Diamond (◇) denotes the mean value of the number of OTUs. Asterisks (**) indicate a significant difference ($p < 0.01$), and NS abbreviates no significant difference.

a maximum of 8, which were above zero for the majority of samples with the exception of five samples (Fig. 5).

From the 92 species detected using the new database, 45 were detected only based on the newly registered sequences (Table 3). 18 species were detected based on several OTUs, when using the new and old databases and of those, 15 species were detected by both new and existing sequences (Table S3). In the newly registered sequences, 13 species had higher (mean $0.005 \pm 0.002$) or equal similarity between the sequences of a singles species, except for two species, *Eucalanus bungii*, and *Metridia lucens*. By the addition of new sequences, the occurrence frequency increased or was the same as for existing sequences (14 species, mean $1.6 \pm 0.8$ samples), except for *Metridia pacifica* in the number of occurrences. The detection sequences also increased for 13 species (mean $215 \pm 266$ sequences) with the exception of *Metridia lucens* and *Metridia pacifica*.

For the Sea of Okhotsk (Okhotsk Tower), 45 species associated with the newly registered sequences were detected. From those, 16 species were detected based on the sequences registered from the Sea of Okhotsk (Table S3). Eight species (*Pseudocalanus newmani* present in 175 samples, *Oithona atlantica* 127 samples, *Oithona similis* 105 samples,

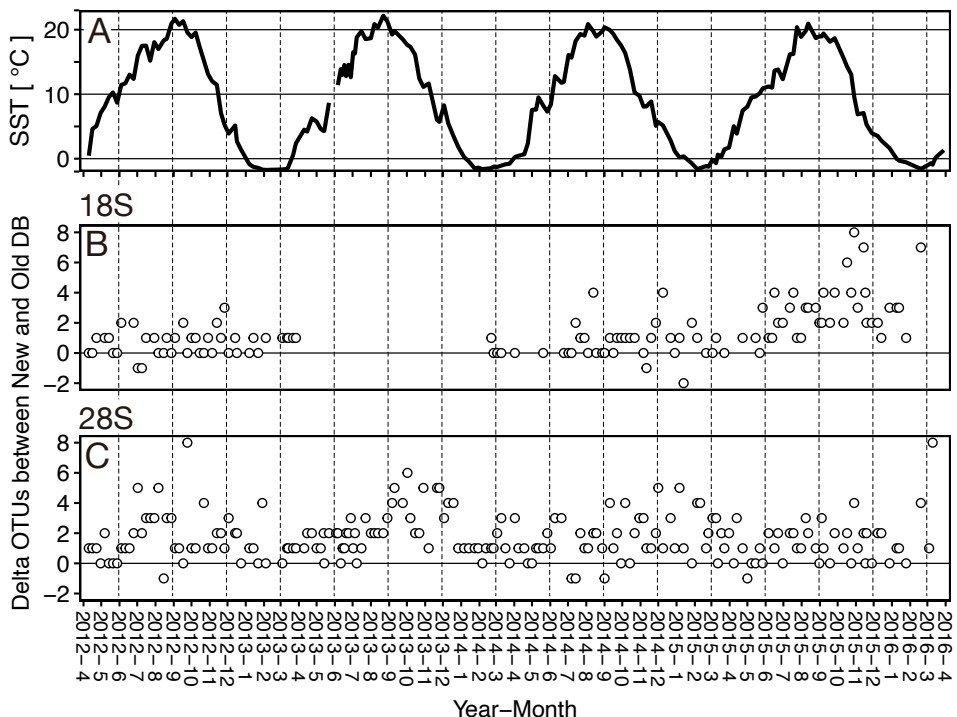

**Figure 5** **Seasonal variation of the sea surface temperature and the difference of (delta) OTUs detected based on the rRNA genes from metabarcoding samples from the Okhostk Tower based on the new and old reference databases.** (A) Seasonal variation of the sea surface temperature (SST). Seasonal variation of delta OTUs based on the 18S rRNA gene (B) and based on the 28S rRNA gene (C). The delta OTUs were obtained by subtracting the number of OTUs in the old database from the number of OTUs in the new database. The delta OTUs greater than 0 indicates that the number of OTUs in the New database is larger than in the Old database. The horizontal axis indicates the sampling period (10 April 2012 to 29 March 2016). The vertical dotted lines separate the seasons every three months. All data is shown in Table S2.

*Pseudocalanus mimus* 103 samples, *Mesocalanus tenuicornis* 100 samples, *Neocalanus flemingeri* 71 samples, *Eucalanus bungii* 39 samples, *Microsetella norvegica* 39 samples) displayed high occurrence frequencies (>20% in >35 samples out of 194 samples). The remaining 29 detected species were registered from other areas, including 27 species of Copepoda, one species of Branchiopoda, and one species of Sagittoidea. Six species(*Clausocalanus arcuicornis* present in 80 samples, *Clausocalanus pergens* 74 samples, *Centropages abdominalis* 69 samples, *Ditrichocorycaeus affinis* 53 samples, *Clausocalanus farrani* 47 samples, *Oncaea venusta* 43 samples) that displayed high occurrence frequencies >20% in >35 samples, identified based on sequences registered from the CTWNP and/or the Sea of Japan. Among these, 13 species of Copepoda (*Acartia danae, Acartia negligens, Calanus agulhensis, Scolecithricella dentata, Oithona longispina, Oithona nana, Oithona pulla, Oithona setigera, Oncaea bispinosa, Oncaea venusta venella, Oncaea zernovi, Macrosetella gracilis, Euterpina acutifrons*) were detected based on the sequences registered from warm waters, such as the WTWNP, East China Sea, and the Nansei Islands.

## Relationship between seasonality and inclusion of sequences from several areas

At the sampling location (Okhotsk Tower) in the Sea of Okhotsk, the sea surface temperature (SST) showed clear seasonality with low temperatures in winter (December–January)/spring (March–May) and high temperatures in summer (June–August)/autumn (September–November), and the average temperatures in each season were $3.8 \pm 4.0$ °C for spring, $16.4 \pm 3.4$ °C for summer, $13.8 \pm 5.4$ °C for autumn, and $0.8 \pm 2.6$ °C for winter (Fig. 5, Table S2). Based on the 18S marker, the delta OTUs (OTUs of the new database minus the old database) increased from the summer to autumn of 2015 (Fig. 5). The mean of delta OTUs in each season was: $0.6 \pm 0.7$ in spring, $1.3 \pm 1.4$ in summer, $1.9 \pm 2.0$ in autumn, $1.3 \pm 1.8$ in winter (Fig. 6). There was a significant difference in Delta OTUs between spring and autumn ($p < 0.01$). Based on the 28S marker, the delta OTUs showed an increasing trend in the autumn (Fig. 5). The mean of delta OTUs detected in each season was $1.2 \pm 1.4$ in spring, $1.7 \pm 1.3$ in summer, $2.1 \pm 1.9$ in autumn, and $1.8 \pm 1.5$ in winter (Fig. 6). There was a significant difference in the Delta OTU between spring and summer, and spring and autumn ($p < 0.01$).

## DISCUSSION

The variable ocean environment is one of the main factors supporting marine diversity including zooplankton in Japanese waters, and the molecular approach is a useful tool to solve difficulties in species identification (*Fujikura et al., 2010*; *Ohtsuka & Nishida, 2017*). In this study, we focused on dominant zooplankton species collected from six areas around Japan covering subarctic to subtropical regions. Taxonomically verified sequences of both 18S and 28S rRNA genes were obtained for nearly 100 species belonging to Arthropoda and Chaetognatha. Among those species about 66 and 41 species were previously not represented in the public databases based on the 18S and 28S rRNA gene sequences. This indicates that there were many unregistered rRNA gene sequences even for the dominant zooplankton species in the western North Pacific Ocean. Based on a test with metabarcoding data from field samples, the taxonomically verified and newly registered sequences contributed to the improved taxonomic classification. More than 19 species detected based on both the 18S and 28S markers were taxonomically identified based on the newly registered sequences, including the dominant zooplankton species (*e.g.*, *P. newmani*). The accuracy of metabarcoding analysis largely depends on the sequence data using specimens classified by taxonomic specialists (*Lindeque et al., 2013*). Based on the results of this study we further demonstrated the importance of registration of taxonomically verified sequences for zooplankton to improve identification in metabarcoding analysis from the marine waters around Japan.

In this study, a large number of the taxonomically verified and newly registered sequences were from the small copepods belonging to *Oithona* and *Oncaea* based on both the 18S and 28S markers, and *Clausocalanus* based on the 18S marker. The sequences associated with those genera were mainly registered from warm sea areas of the WTWNP and the East China Sea. The sequence data registration has previously mainly focused on marine zooplankton

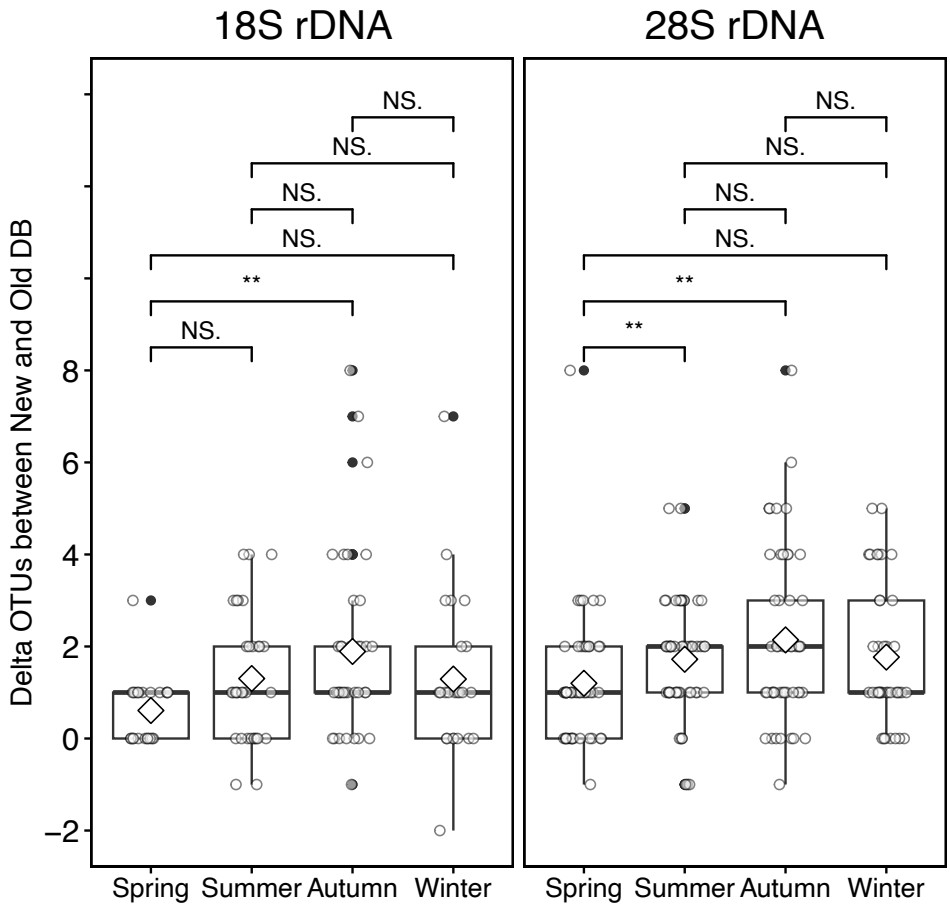

**Figure 6** **Boxplot of the difference (delta) of OTUs from rRNA genes metabarcoding samples between new and old reference databases by seasons at the Okhotsk Tower.** The delta OTUs from 18S (left) and 28S (right) rRNA gene sequences were compared between seasons. The seasons were separated by spring, March-May; summer, June-August; autumn, September-November; winter, December-February. Diamond ( ◇ ) indicates the mean value of the delta OTUs by season. Asterisks (**) indicate a significant difference ($p < 0.01$), and NS abbreviates no significant difference.

from high latitudes with low water temperatures (*Bucklin et al., 2010a*). At the same time, high zooplankton diversity is observed in the low latitudes with warm water temperatures, where the community is mostly composed of small zooplankton species (*Turner, 2004*). In addition to species in *Oithona, Oncaea,* and *Clausocalanus*, other small copepod species were detected based on newly registered sequences including species from the family Acartiidae, Clausocalanidae, Paracalanidae, and Corycaeidae (*Gallienne & Robins, 2001; Turner, 2004*). The importance of small zooplankton, in particular non-calanoid copepods including *Oithona* and *Oncaea*, has been previously overlooked (*Böttger-Schnack, 1995; Turner, 2004*), and we found that sequence data from these small copepods were also limited in the public database. The metabarcoding approach is efficient for detecting diversity, including small copepods, and in this study morphologically verified sequence registration

aided successful identification of small species in metabarcoding analyses targeting the 18S and 28S rRNA genes.

Although the number of available 18S rRNA gene sequences is high for eukaryotic taxa, our study showed higher proportions of species with newly registered sequences for the 18S than for the 28S rRNA gene. Although both the 18S and 28S rRNA genes have been used for phylogenetic analysis of marine copepods (*Blanco-Bercial, Bradford-Grieve & Bucklin, 2011*), the 28S rRNA gene has been commonly used for species identification (*Kiesling et al., 2002*; *Blanco-Bercial, Bradford-Grieve & Bucklin, 2011*). For example in the Kuroshio region in the western North Pacific, sequences for more than 100 calanoid copepod species have been registered based on the 28S rRNA gene (*Hirai et al., 2015*). The metabarcoding analysis also showed larger numbers of OTUs with BLAST-hit to newly-registered sequences based on the 18S marker than by the 28S marker, supporting the importance of adding taxonomically verified sequences, especially for the 18S rRNA gene, genes to public databases. The species-specific homology of zooplankton sequences showed higher values for newly registered sequences than for the existing sequences for many species, and the frequency of occurrence and the number of sequences also increased. This suggests that the registration of reference sequences of zooplankton species found from the field, which may differ between individuals from different sampling location, has improved identification accuracy and detection sensitivity.

In the metabarcoding analysis based on the filed samples from the Sea of Okhotsk, the number of detected OTUs increased based on the new database containing the newly registered sequences. The different OTUs of the same species were detected based on the old and new databases. It was suggested that the same reads in metabarcoding samples were detected as a different OTU due to registering new reference sequences with higher similarity. This result shows improved species identification not only at species level but also at OTU level by the new sequence registration. The seasonal differences of OTUs representing a single species with BLAST-hit to newly registered sequences differed between the 18S and 28S rRNA genes. The number of OTUs identified based on the newly registered sequences was relatively high based on the 18S and 28S rRNA genes in autumn. This can be explained by the strong seasonal variability in the dominant currents present in the sampling location. The Soya Warm Current (SWC) is especially dominant in autumn on the east coast of Hokkaido in the Okhotsk Sea, and warm water zooplankton species are transported by the SWC from the Sea of Japan (*Hamaoka et al., 2010*). In the metabarcoding analysis, we also detected warm water species such as *C. elliptica*, *T. gracilis*, *O. aruensis*, for which the taxonomically verified sequences originated only from the Nansei Islands. Because about half of the species detected from the Sea of Okhotsk based on the newly registered sequences originating from other sea areas, the accuracy of species identification can be improved by including zooplankton sequences not only from the survey area but also from other sea areas. The increase in the number of zooplankton species that could be detected by the taxonomically verified sequences added to the database in this study shows that further registration of taxonomically verified sequences is important.

There is also a need for the registration of rare (non-dominant) zooplankton species for further improvement of taxonomic identification based on molecular techniques.

For example, the 18S rRNA gene is relatively conserved, and the same sequences are observed between closely related species (*Tang et al., 2012*). When only one species was registered between two closely related species, the other unregistered species can be misidentified by metabarcoding analysis. In addition, combinations with other molecular markers with high mutation rates (*e.g.*, mitochondrial COI) would be useful for solving problems of taxonomic resolution, although these markers have difficulty in amplifying various zooplankton species with high diversity. We focused on the rRNA genes, which are common for eukaryotic metabarcoding; however, registration of other markers is also helpful to improve the accuracy of metabarcoding using multiple markers (*Neigel, Domingo & Stake, 2007*). Because molecular tools are useful for the taxonomic characterization of zooplankton communities, we hope that future efforts of sequence registration will support further understanding of the roles of zooplankton in the marine ecosystem and facilitate the detection of long-term changes in their community structure.

## CONCLUSIONS

This study aimed to improve zooplankton species identification based on metabarcoding by registering taxonomically verified sequences from six environmentally different areas in Japanese waters. A total of 331 18S and 28S rRNA gene sequences from 100 zooplankton species were registered to a public database. In the field samples used for investigating the improvement of taxonomic identification, even the dominant species lacked the sequences available in the public database. Those species with unregistered sequences were mainly composed of small non-calanoid copepods (*i.e., Oithona* and *Oncaea*). Verified improvement of taxonomic identification was demonstrated by more than one-fifth of OTUs from the field samples that were identified to species level based on the newly registered sequences of the 18S and 28S rRNA genes. The reference database including the taxonomically verified sequences improved the accuracy of zooplankton detection both overall and in individual samples, and the accuracy of species identification also improved at the population level due to higher similarity values among sequences of the same species. Continuous registration of sequence data covering various environmental conditions is necessary for further improvement of taxonomic identification of zooplankton using metabarcoding analysis and it also facilitates monitoring of marine ecosystems by molecular tools.

## ACKNOWLEDGEMENTS

K. Kamiya and S. Ori are thanked for the technical assistance. We thank H. Yoshida and K. Murai for the plankton sampling and CTD observation around the Okhotsk Tower. We thank the captains, officers, crews, and scientists of R/V Hokko-Maru, R/V Wakataka-Maru, R/V Shunyo-Maru, R/V Tenyo-Maru, R/V Soyo-Maru, R/V Yoko-Maru, and the Fisheries Agency of Japan, for their invaluable support with sampling. We are grateful to Dr. Y. Igeta for the management of sampling in the Sea of Japan. We extend our gratitude to Drs. H. Kuroda and T. Nakanowatari for collecting the samples.

### Funding

This work was supported by "Establishing a network of environment and fisheries information", Ministry of Agriculture, Forestry and Fisheries, Japan. The funders had no role in study design, data collection and analysis, decision to publish, or preparation of the manuscript.

### Grant Disclosures

The following grant information was disclosed by the authors:
"Establishing a network of environment and fisheries information", Ministry of Agriculture, Forestry and Fisheries, Japan.

### Competing Interests

Noriko Nishi is employed by AXIOHELIX Co. Ltd.

### Author Contributions

- Tsuyoshi Watanabe conceived and designed the experiments, analyzed the data, prepared figures and/or tables, authored or reviewed drafts of the article, and approved the final draft.
- Junya Hirai conceived and designed the experiments, analyzed the data, authored or reviewed drafts of the article, and approved the final draft.
- Sirje Sildever conceived and designed the experiments, analyzed the data, authored or reviewed drafts of the article, and approved the final draft.
- Kazuaki Tadokoro conceived and designed the experiments, performed the experiments, analyzed the data, prepared figures and/or tables, authored or reviewed drafts of the article, and approved the final draft.
- Kiyotaka Hidaka conceived and designed the experiments, performed the experiments, analyzed the data, prepared figures and/or tables, authored or reviewed drafts of the article, and approved the final draft.
- Iwao Tanita conceived and designed the experiments, performed the experiments, analyzed the data, prepared figures and/or tables, authored or reviewed drafts of the article, and approved the final draft.
- Koh Nishiuchi conceived and designed the experiments, performed the experiments, analyzed the data, prepared figures and/or tables, authored or reviewed drafts of the article, and approved the final draft.
- Naoki Iguchi conceived and designed the experiments, performed the experiments, analyzed the data, prepared figures and/or tables, authored or reviewed drafts of the article, and approved the final draft.
- Hiromi Kasai conceived and designed the experiments, performed the experiments, analyzed the data, prepared figures and/or tables, authored or reviewed drafts of the article, and approved the final draft.

- Noriko Nishi conceived and designed the experiments, performed the experiments, analyzed the data, authored or reviewed drafts of the article, and approved the final draft.
- Seiji Katakura conceived and designed the experiments, performed the experiments, analyzed the data, authored or reviewed drafts of the article, and approved the final draft.
- Yukiko Taniuchi conceived and designed the experiments, performed the experiments, analyzed the data, authored or reviewed drafts of the article, and approved the final draft.
- Taketoshi Kodama conceived and designed the experiments, performed the experiments, analyzed the data, authored or reviewed drafts of the article, and approved the final draft.
- Satokuni Tashiro conceived and designed the experiments, performed the experiments, analyzed the data, authored or reviewed drafts of the article, and approved the final draft.
- Misato Nakae conceived and designed the experiments, performed the experiments, analyzed the data, authored or reviewed drafts of the article, and approved the final draft.
- Yuji Okazaki conceived and designed the experiments, performed the experiments, analyzed the data, authored or reviewed drafts of the article, and approved the final draft.
- Satoshi Kitajima conceived and designed the experiments, performed the experiments, analyzed the data, authored or reviewed drafts of the article, and approved the final draft.
- Sayaka Sogawa conceived and designed the experiments, performed the experiments, analyzed the data, authored or reviewed drafts of the article, and approved the final draft.
- Toru Hasegawa conceived and designed the experiments, performed the experiments, analyzed the data, authored or reviewed drafts of the article, and approved the final draft.
- Tomonori Azumaya performed the experiments, analyzed the data, authored or reviewed drafts of the article, and approved the final draft.
- Yutaka Hiroe performed the experiments, analyzed the data, authored or reviewed drafts of the article, and approved the final draft.
- Daisuke Ambe performed the experiments, analyzed the data, authored or reviewed drafts of the article, and approved the final draft.
- Takashi Setou performed the experiments, analyzed the data, authored or reviewed drafts of the article, and approved the final draft.
- Daiki Ito performed the experiments, analyzed the data, authored or reviewed drafts of the article, and approved the final draft.
- Akira Kusaka performed the experiments, analyzed the data, authored or reviewed drafts of the article, and approved the final draft.
- Takeshi Okunishi performed the experiments, analyzed the data, authored or reviewed drafts of the article, and approved the final draft.

- Takahiro Tanaka performed the experiments, analyzed the data, authored or reviewed drafts of the article, and approved the final draft.
- Akira Kuwata performed the experiments, analyzed the data, authored or reviewed drafts of the article, and approved the final draft.
- Daisuke Hasegawa performed the experiments, analyzed the data, authored or reviewed drafts of the article, and approved the final draft.
- Shigeho Kakehi performed the experiments, analyzed the data, authored or reviewed drafts of the article, and approved the final draft.
- Yugo Shimizu performed the experiments, analyzed the data, authored or reviewed drafts of the article, and approved the final draft.
- Satoshi Nagai conceived and designed the experiments, performed the experiments, analyzed the data, prepared figures and/or tables, authored or reviewed drafts of the article, and approved the final draft.

## DNA Deposition

The following information was supplied regarding the deposition of DNA sequences:

The 18S and 28S rDNA sequences are available at the DDBJ databank: LC581890 to LC582220).

## Data Availability

The data is available at DDBJ and GenBank: DRP009697; PRJDB11340; LC581890–LC582220.

## Supplemental Information

Supplemental information for this article can be found online at http://dx.doi.org/10.7717/peerj.15427#supplemental-information.

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
