# Peer review of "Improving taxonomic classification of marine zooplankton by molecular approach: registration of taxonomically verified 18S and 28S rRNA gene sequences"

_PeerJ, doi:10.7717/peerj.15427_

## Round 0.1 · original submission · Major Revisions

The two reviewers at hand (a third one failed to provide his/her review on time) concur that your work is interesting but were also critical on several issues, especially Reviewer #2. Concerns about the methodology raised also by Reviewer #1 are important and may compromise the conclusions reached in your study. Please consider carefully the remarks about seasonality, which currently appears as a somewhat disconnected section.

A problem with English usage was found by Reviewer #2, with whom I concur. S/he has picked a few examples but some more can be found in the manuscript. Please pay attention to their suggested changes and comments for improvement and have your manuscript thoroughly revised regarding grammar and syntax. I am looking forward to receiving your revised version.

·

Basic reporting

The manuscript is clear and well written. I have some minor comments/questions raised to the authors, to clarify.

Experimental design

Pre.cluster with 4 diffs is a lot (would preclusters sequences with up to 8 bp differences). Why was needed such high number of bp, considering it was 18S and 28S? Maybe due to the long fragments? Illumina quality suffers when fragments are >400bp (since the overlapping regions for R1 and R2 are then low quality for both threads).

And then clustered to 0.99. But, if ribosomal sequences for such short fragments are often shared between species of the same genus (even D1D2) why clustering to 99%? That would clump into a single OTU several species for 18S, and likely for 28S too. I wonder if choosing such long fragments, with today's technology, would have a negative effect in our ability to detect species since we have to then precluster and cluster. But, it might be positive if long sequences improves.

Regarding taxonomic identification of the OTUs: which was the criteria when a blast search had multiple hits, with the same bitscore, but to different species? At least, with 18S that is a very common occurrence.

Validity of the findings

Table S1 it is very interesting – showing when the markers are not species-specific. Which proportion do the authors think is due to misID in the databases, and which one due to real shared sequence? (in 28S, I suppose). The high number to 28S to genus level is surprising. Or - does just it mean there was no species in Genbank.

Have the authors calculated how many species had a unique 18S, how many unique 28S, and how many share any of them? (in the barcoding effort).

Out of curiosity – when sequencing six colonies of the same individual – did the colonies agreed in the same sequence, or did they differ from each other? If they differed, which was the criteria to pick one? Something to think about – including them all in the DB or not!

Additional comments

Supplemental table 4 is cited before ST 2 and ST 3. Please renumber accordingly (that is, 4 should be 2, 2 should be 3 and 3 should be 4). Also, please check the name of the files, since they are S1, S2, S4 and S5.

Reviewer 2 ·

Basic reporting

The level of English of the manuscript hinders the comprehension of its content. I think it needs to be improved to make the text understandable. Incorrect sentences can be found along the text, e.g., in lines 131 (“The aims to…”), 154-155 (“was followed by the worms”), 183 (“with a reaction mixture consisted of…”), 247-248 (“the second PCR was performed in the reaction mixture volume was 50 ul with”), 401-402 (“the numbers of species 26±17 species were detected per sample”), etc. Also, in some cases the authors confound the definite article “the” with the indefinite article “a”. Some examples of this issue can be found in line 172 (“The alignment was done…”) or 203 (“The BLAST search”).
The structure of the manuscript follows a usual research format and includes all standard sections expected in this type of documents.
The tables are quite confusing for me, it is difficult to understand the information that is contained in. Also, the table captions need to be improved to help understand the information the tables contain. It also applies to Supplementary Tables.
I am not sure if Figure3 is needed.
I did not find any link nor mention to raw data in the text.
From my point of view, the manuscript is self-contained, and represents an appropriate unit of publication. Yet, I think that there is no need to separate in different subsections the results of “registration of new sequences” and “improvement of taxonomic classification” for 18S and 28S genes, since it makes the text repetitive. I would suggest two subsections (“registration…” and “improvement…”) showing the results for both barcodes. Also, I would suggest excluding the seasonality results from the paper or to place it in a different subsection in the results section.

Experimental design

The work by Watanabe and collaborators fits within the Aims and Scope of the journal.
The study clearly defines the research question, which is related to the importance of having complete reference databases for a successful application of molecular methods (mainly DNA metabarcoding) for monitoring of marine ecosystems. From my point of view, the topic is timely and relevant (particularly for molecular ecologists). The main question is clearly defined, and the authors clearly state how they findings contribute to address the question.
Yet, I think that the hypothesis of the manuscript should be reformulated and placed into a more current context. My arguments: the hypothesis that increasing reference databases improves the accuracy of the taxonomic classification in DNA metabarcoding has been already tested and demonstrated in other groups (for instance, see Gold et al. 2021 doi: 10.1111/1755-0998.13450, or Collins et al. 2021 doi: 10.1111/jfb.14852). It is important to note that the research question addressed by the authors is purely methodological and specific to DNA metabarcoding. Thus, the methodological findings in one taxonomic group also apply to the others when using this molecular technique. I would recommend the authors to adjust the hypothesis and the focus of the manuscript in order to accommodate it to the current knowledge.
Also, I think that an additional analysis would be needed. This new analysis would consist of assigning taxonomy to the sequences using the same reference database (DB) BUT excluding the new 18S and 28S sequences. This will help to better show the findings of the authors (examples of other works addressing similar issues: Gold et al. 2021 doi: 10.1111/1755-0998.13450; Collins et al. 2021 doi: 10.1111/jfb.14852). This additional analysis will allow the quantification of OTUs that are taxonomically classified using the DB with the new barcoded sequences that remained as unclassified using the DB without the new sequences. It will also report possible misidentifications (OTUs assigned to a species that are assigned to a new one due to lack of the latter in the previous DB). Both topics are of high interest to evaluate the strengths and weaknesses of DNA metabarcoding for monitoring purposes.
I see no major methodological flaws in the study that may question the high technical standards applied in the present work, yet I think the text needs to be more improved since some sentences or paragraphs are ambiguous and/or difficult to understand (the authors will find some specific comments and questions in the “General comments” section).
As far as I know, there are not “ethical standards” in the field of the study more than the ethics related to do “good science”, for which I did not find any evidence questioning it.
I think the methods exposed by the authors are described with sufficient detail to be replicated although, as stated above, I found some paragraphs/sentences a bit confusing, difficult to understand or lacking in detail, so they should be improved to facilitate its understanding (and therefore its replication). See “General comments” section for more details.

Validity of the findings

From my point of view, the findings of the present manuscript are valid, with conclusions that are appropriately stated, connected to the original question, and limited to the results shown and discussed in the study. However, I think that some results shown and discussed do not really address the main question of the manuscript (the seasonality results).

Additional comments

First of all, I would like to remark that it is really nice to see studies that push forward barcoding efforts and give value to the taxonomic knowledge, which is currently decreasing and sometimes neglected. I would like to acknowledge the authors their efforts in the field.
Apart from the comments made above, please find some additional comments below:
Line 46. RNA? I think the authors may refer to DNA or to rRNA.
Line 57. What do you mean with 21/115 OTUs? 21 out of 115? I think it should be properly written. The same in lines 385, 410, 440…
Line 92. Add only the year of the reference in parentheses, the authors are already stated.
Line 131. “The aims to verify this hypothesis using natural zooplankton samples”. Probably “The aim is …” instead of “The aims …”?
Line 150. Instead of saying “or more”, which is inaccurate, I would suggest the authors to give a range of time (minimum-maximum time) until the ethanol was replaced. Example: “ethanol was replaced between 24h and 3 days of initial preservation”.
Lines 182-190. It is not clear for me whether 18S and 28S PCR reactions were carried out in the same reaction tube (multiplex PCR) or independently. This is important and needs to be clarified. Also, it seems that the cycling conditions were the same for both primer pairs (18S and 28S). If this is the case, I would recommend to clearly state it in the text. If it is not the case, then please clearly specify the cycling conditions used for each of the primers. Also, it would be useful to have in the text the longitude of the region amplified by the 28S rRNA primer pairs; the text only says that it was designed to have less than 500 bp, not the actual length of the barcode).
Line 201. The M13 and U19 primers should be referenced or placed into better context. There is no clue in the text where these primers come from.
Line 214. There is no need for “respectively” in the sentence.
Lines 218-219. I think it would be worth to clarify how the authors split the samples for DNA extraction. Did the authors extract half/one third/25% of the sample? Did they extract the DNA of the different portions of the sample and then merged all the extracted DNA into one unique integrated DNA sample?
Line 261. Maybe 5’ instead of 50?
Lines 268-270. This sentence is quite confusing. It is not clear for me what the authors did here.
Lines 277-279. From my point of view, the usage of the term OTU is a bit confusing here. The unique.seqs command does not provide OTUs but dereplicate sequences (i.e., instead of having 20 identical sequences, you will have 1 sequence with a size of 20). I think the term OTU fits better in the clustering step, “sequences were clustered into OTUs at a 0.99 level…”.
Lines 288-289. This sentence is difficult to understand. Please rewrite it in order to make it clearer.
Lines 297-306. It was impossible for me to clearly understand what the authors did here. I think the text should be rewritten to make it easier to follow. Please avoid ambiguous explanations such as the ones in lines 301-303: which errors do the authors refer to? How do the authors know that these errors do not represent molecular diversity? (Especially in the 28S gene, which provides taxonomic resolution to species level.) I think it needs to be further detailed.
Line 321. I am not sure whether “occupied” is correct here.
Line 323. Order Calanoida instead of Calanoid.
Lines 376-377. How can the authors refer to an improvement of taxonomic classification when they have nothing to compare their results with? I see more logic to assign taxonomy to the sequences using i) the reference database with pre-existing sequences, and ii) the reference database adding the new sequences.
Line 396-407 and 431-435. I am not sure about what seasonality adds to the work. I think it is not needed for answering the main question of the authors. However, in case the authors want to maintain it, it should have its own subsection.
Figure 1.
- I would recommend making more visible where the zoom of Nansei Islands comes from. It was not easy for me to find it.
- Figure 1 caption: write the full name of “Warm Temperate WNP” as you did with the Cold.
- Figure 1 caption: It should be clearly stated in the figure caption that Okhotsk Tower is the station in which metabarcoding samples were collected.
Figure 3. Is this figure really needed? The information it shows is clearly stated in the text, I think there is no need for a piechart only for this data.

---

## Round 0.2 · accepted · Accept

Your manuscript has been reviewed by the previous reviewers and both agree that you have addressed their comments thoroughly, substantially improving its readability and overall quality. I wholly concur with their opinion, hence your manuscript is ready for publication.

·

Basic reporting

The authors have significantly improved the manuscript, and answered all my questions.

I have no further concerns regarding this manuscript.

Experimental design

The methods are now clearly explained.

Validity of the findings

The findings and discussion are adequate to the scope of the manuscript.

Reviewer 2 ·

Basic reporting

The authors did a great job improving the manuscript. It reads now much more clear and unambiguous. They have also addressed all my doubts and suggestions.

Experimental design

The authors have improved substantially the work by adding the requested additional analysis, and by notably improving the "material and methods" section.

Validity of the findings

I was not sure about including the seasonality results in the manuscript (I am not sure yet). However, I acknowledge that the new focus of the "seasonality" section fits better to the scope of the manuscript.

Additional comments

I have no additional comments.